# Managing Super Pests: Interplay between Pathogens and Symbionts Informs Biocontrol of Whiteflies

**DOI:** 10.3390/microorganisms12050887

**Published:** 2024-04-28

**Authors:** Weili Yan, Saixian Wang, Jialei Liu, Dan Zhai, Hang Lu, Jingjing Li, Rune Bai, Caiyan Lei, Luyang Song, Chenchen Zhao, Fengming Yan

**Affiliations:** Henan International Laboratory for Green Pest Control, College of Plant Protection, Henan Agricultural University, Zhengzhou 450046, China; weiliyan0826@163.com (W.Y.); saixian0828@163.com (S.W.); liujialei2022@163.com (J.L.); z13598789312@163.com (D.Z.); lhang_990912@163.com (H.L.); jjli@henau.edu.cn (J.L.); bairune@henau.edu.cn (R.B.); leicaiyanlcy@126.com (C.L.); lysong@henau.edu.cn (L.S.)

**Keywords:** non-viruliferous versus viruliferous whitefly, *Metarhizium anisopliae*, pathogenic, symbiotic bacteria, insect–plant–virus interaction

## Abstract

*Bemisia tabaci* is distributed globally and incurs considerable economic and ecological costs as an agricultural pest and viral vector. The entomopathogenic fungus *Metarhizium anisopliae* has been known for its insecticidal activity, but its impacts on whiteflies are understudied. We investigated how infection with the semi-persistently transmitted Cucurbit chlorotic yellows virus (CCYV) affects whitefly susceptibility to *M. anisopliae* exposure. We discovered that viruliferous whiteflies exhibited increased mortality when fungus infection was present compared to non-viruliferous insects. High throughput 16S rRNA sequencing also revealed significant alterations of the whitefly bacterial microbiome diversity and structure due to both CCYV and fungal presence. Specifically, the obligate symbiont *Portiera* decreased in relative abundance in viruliferous whiteflies exposed to *M. anisopliae*. Facultative *Hamiltonella* and *Rickettsia* symbionts exhibited variability across groups but dominated in fungus-treated non-viruliferous whiteflies. Our results illuminate triangular interplay between pest insects, their pathogens, and symbionts—dynamics which can inform integrated management strategies leveraging biopesticides This work underscores the promise of *M. anisopliae* for sustainable whitefly control while laying the groundwork for elucidating mechanisms behind microbe-mediated shifts in vector competence.

## 1. Introduction

Plant viruses pose significant threats to global agriculture by reducing crop yield and quality. These pathogens can lead to devastating economic losses and food security challenges [1]. Among the approximately 1100 recognized plant viruses, a substantial 80% are transmitted by insect vectors, predominantly those with piercing and sucking feeding behaviors, such as whiteflies, aphids, and planthoppers [2]. Effective management of these vectorial insects stands as a potent strategy to mitigate the prevalence of plant viral diseases [3]. Consequently, vigilant surveillance and timely intervention targeting these insect vectors are paramount to thwart the proliferation of associated diseases [4]. Furthermore, an ongoing evolutionary contest ensues between plant viruses and their respective hosts, which inevitably shapes the distribution and behavioral patterns of pest populations [5,6,7].

*Bemisia tabaci* Gennadius (Hemiptera: Aleyrodidae), globally recognized as a super pest, plays a pivotal role in the dissemination of over 400 plant viruses [8,9]. The intricacies of the *B. tabaci* complex are vast, comprising over 46 cryptic species, and have been implicated in substantial reductions in crop yields. Notably, this complex serves as a vector for in excess of 100 begomovirus species, thereby placing numerous global crop cultivations in jeopardy [10]. Historically, the predominant modus operandi for insect pest management has been the deployment of synthetic chemical insecticides. The heavy reliance on chemical pesticides has unintentionally caused environmental pollution, health risks for humans, and threats to biodiversity. It has also led to insects developing resistance to these pesticides [11]. Despite using many different synthetic insecticides to control whiteflies, the continuous use of neonicotinoids has led to increased resistance. This makes managing the pest more difficult. Interestingly, Yan et al. found a link between a whitefly’s ability to transmit viruses and its resistance to imidacloprid, a common neonicotinoid [12].

The effective management of agricultural pests, such as *B. tabaci*, remains a critical challenge due to the extensive use of synthetic insecticides. Consequently, there is an urgent need for alternative strategies that minimize the negative impacts of insecticides while effectively controlling pest populations. Entomopathogenic fungi (EPF) like *Metarhizium anisopliae* Metsch (Hypocreales: Clavicipitaceae) offer a promising solution due to their specific pathogenicity to insects [13]. Unlike synthetic insecticides, EPFs provide a biologically-based approach to pest control that can be integrated into sustainable agricultural practices [14]. The effectiveness of EPFs, particularly at sublethal concentrations, has opened new avenues for pest management by not just killing pests outright but by potentially affecting their physiological and reproductive fitness.

Sublethal effects, such as alterations in pest behavior, fecundity, and lifespan, may reduce pest populations and their impact on crops more subtly but effectively. These effects include behavioral changes that decrease feeding, reduce mating success, and impair mobility, ultimately contributing to lower population viability [15]. In addition to these direct impacts, *M. anisopliae* may also affect the microbiota of insects, which can have profound secondary effects on their fitness and vector capacity.

We hypothesize that *M. anisopliae* negatively affects the fitness of whiteflies through both the direct pathogenic effects and the subtle sublethal influences it has on the insect host. By investigating the interactions between whiteflies and their microbial communities under the influence of *M. anisopliae*, we aim to shed light on a less-understood aspect of biocontrol—the role of microbiota in pest management. The insights gleaned from this investigation are pivotal, not only in illuminating the intricate dynamics of plant–insect-insecticide interactions in the presence of this pathogenic fungus but also in augmenting the strategic deployment of *M. anisopliae* within an integrated framework for whitefly management. Understanding these interactions provides crucial insights into the holistic management of whiteflies and potentially other pest species, offering a sustainable complement or alternative to traditional chemical control methods.

## 2. Materials and Methods

### 2.1. Plants, Insects, Virus, and Fungus

Plants (cucumber, cotton), whiteflies, and CCYV were procured from sources previously reported by Yan et al. [12]. The plants were grown in pots of 10 cm in diameter and 12 cm in height, contained within insect-proof enclosures measuring 60 × 40 × 80 cm. These enclosures were situated in a controlled laboratory environment, where the temperature was maintained at 28 ± 0.5 °C, the relative humidity was kept at 65 ± 5%, and the light/dark cycle was set to 16 h of light followed by 8 h of darkness. The rearing of the *B. tabaci* Mediterranean (MED; biotype Q) cryptic species was conducted independently at Henan Agricultural University, where it was propagated on cucumber plants for a duration of 5 years.

The colonies of non-viruliferous whitefly and viruliferous whitefly were sustained on cucumber and cotton plants over 50 generations free from insecticide exposure before this experiment. Non-viruliferous *B. tabaci* adults were collected from healthy plants. Non-viruliferous *B. tabaci* individuals reached to 100% acquisition of CCYV at 48 h after feeding at 48 h after feeding on CCYV-infected cucumber plants [16]. We determined the viruliferous status of whiteflies using real-time RT-PCR [16,17].

CCYV-infected plants (cucumber) were generously furnished by Prof. Xiaobin Shi from Hunan Academy of Agricultural Sciences. These cucumber plants played a pivotal role in both the maintenance of the virus clone and transmission assays. Cotton, whilst being an amenable host for whiteflies, remains unsuitable for CCYV. Cotton was used to test how long whiteflies could still transmit the virus after being removed from CCYV-host plants. Whiteflies carrying CCYV were first allowed to feed on CCYV-infected plants. Then, they were moved to healthy, non-infected cucumber plants. This ensured the whiteflies effectively transmitted the virus.

The entomopathogenic fungus *M. anisopliae* (strain IBCCM321.93) was cultured on potato dextrose agar for a fortnight in an incubator set at 25 ± 1 °C, 80 ± 0.5% RH, under continuous darkness. Conidia were harvested in a 0.05% Tween 80 solution and refrigerated at 4 °C, remaining viable for 3 to 4 weeks [18]. For experimental purposes, only conidia exhibiting a germination rate exceeding 90% were utilized.

### 2.2. Metarhizium Anisopliae Bioassay of Non-Viruliferous and Viruliferous Whiteflies

Whitefly individuals (non-viruliferous and viruliferous) were chosen for the *M. anisopliae* bioassay, which was conducted by the leaf (cucumber leaf) dip method [11], containing 100 μL of the conidial suspension (10^8^, 10^7^, 10^6^, 10^5^, and 10^4^ conidia/mL, separately) of *M. anisopliae* or sterilized tap water (with 0.05% Tween 80, control) to soak for 10 s, respectively. Whitefly mortality was assessed at 120 h (All insects died) by quantifying the deceased insects to calculate LC_50_ (the concentration needed to kill 50% of the adults) and LT_50_ (the time needed to kill 50% of the adults) (Figure 1A). Each concentration was replicated five times, with each replicate containing 20 individual whiteflies.

### 2.3. Effect of M. anisopliae on the Symbiotic Bacteria to Non-Viruliferous and Viruliferous Whiteflies

To assess the impact of *M. anisopliae* on whiteflies, three-day-old adult non-viruliferous and viruliferous whiteflies were separately introduced to cucumber plants. The cucumber plants were individually treated with specific concentrations of *M. anisopliae* (LC_20_, LC_30_, and LC_50_ of non-viruliferous and viruliferous whitefly, respectively). The *M. anisopliae* solution was uniformly applied to the cucumber leaves using a manual sprayer, and 0.05% Tween 80 was sprayed as a control. After 24 h, whitefly adults were collected rapidly, frozen in liquid nitrogen, and preserved at −80 °C for subsequent analysis (DNA extraction) (Figure 1B). We tested each concentration in six replicates, with each replicate consisting of 100 individual insects.

The microbiota composition of 48 whitefly individuals was characterized using MiSeq sequencing targeting the V3–V4 region of the bacterial 16S gene. Samples derived from whiteflies exposed to *M. anisopliae*-infected and uninfected diets 24 h post-feeding were prepared for analysis. Genomic DNA was isolated from these samples using the MolPure Cell/Tissue DNA Kit (Yeasen Biotechnology, Shanghai Co., Ltd., Shanghai, China), following the provided protocol. Surface sterilization of the whiteflies was achieved through triple rinsing in 75% ethanol and sterile water prior to DNA extraction. Lysozyme treatment (50 mg/mL, Vazyme Biotech Co., Ltd., Nanjing, China) facilitated the lysis of Gram-positive bacterial cells, incubating for 30 min at 37 °C. DNA quality and quantity assessments were conducted using a NanoDrop 2000C spectrophotometer (Thermo Scientific, Waltham, MA, USA) and agarose gel electrophoresis. Amplification of the 16S rRNA gene’s V3–V4 region utilized 338F/806R primers [19], with subsequent PCR product handling, library construction, and sequencing performed as per established protocols [20]. Sequencing was executed on an Illumina MiSeq platform (Illumina, San Diego, CA, USA), with equimolar pooling of the purified amplicons for paired-end sequencing. The generated sequences were deposited in the NCBI Sequence Read Archive under accession number PRJNA1019088.

Data was analyzed as described in previous studies [21], with unique barcodes segregating paired-end reads per library, followed by trimming to obtain standard sequences. Using QIIME [22], the sequences were processed, while Mothur [23] facilitated the filtration based on quality and length criteria. The microbial community was delineated into operational taxonomic units (OTUs) at a 97% similarity threshold, analyzed via Uparse software (v7.0.1001, http://drive5.com/uparse/) [24]. To mitigate sample sequence number variability, rarefied OTU tables were produced. Community richness and diversity were quantitatively assessed using Chao1, Ace, Shannon, and Simpson indices. Comparative analysis of the microbial communities employed Venn diagrams for OTU overlap visualization, Bray-Curtis distance matrices through PCoA, UniFrac metrics for diversity comparisons [25], and nonparametric multivariate analysis of variance via the *R* vegan package for community composition differentiation [26].

### 2.4. Statistical Analyses

The LC_50_ value, toxicity regression equation, and coefficient of determination (R^2^) were determined utilizing the Probit function in IBM SPSS Statistics 20.0. Results were presented as mean ± standard error (SE). Analysis of variance in dominant bacterial populations employed one-way ANOVA, with subsequent Tukey’s honestly significant difference (HSD) test, and two-way ANOVA, executed in SAS software. For non-normally distributed data, the Kruskal–Wallis test was applied (*p* < 0.05). These statistical evaluations were conducted using SAS 9.4 (SAS Institute Inc., Cary, NC, USA).

## 3. Results

### 3.1. Metarhizium Anisopliae Pathogenicity against Whitefly

To determine the pathogenicity of *M. anisopliae* on whitefly, we detected the susceptibility of the fungus-exposed whitefly. We found that the viruliferous whitefly exhibited increased susceptibility to the fungus compared to that of the non-viruliferous whitefly under the same concentration [LT_50_ of 10^8^ conidia/mL, 3.4 d (3.2 d–3.6 d) vs. 2.7 d (2.5 d–2.9 d), χ^2^ = 6.704, *p* = 0.01, Appendix A]. Interestingly, in the control group, viruliferous whitefly showed stronger viability than non-viruliferous whitefly [LT_50_, 5.2 d (4.6 d–6.4 d) vs. 4.7 d (4.5 d–4.9 d), Appendix A].

*M. anisopliae* caused the high adult mortality of all whiteflies tested. With the exception of control, complete mortality was observed in all whiteflies that directly ingested the diet at certain concentrations after five days (Figure 2). Viruliferous whitefly exposed to *M. anisopliae* showed significantly decreased survival compared to the non-viruliferous whitefly (Figure 2A,B). Meanwhile, the control groups exhibited contrary results in terms of survival rate. The viruliferous whitefly were more sensitive to *M. anisopliae* than were the non-viruliferous whitefly. The LC_50_ of *M. anisopliae* for the viruliferous whitefly in 120 h (5 d) ranged from 46,620 to 491,603 conidia/mL diet (Figure 2D). Specifically, the LC_50_ of *M. anisopliae* for the non-viruliferous whitefly ranged from 830,182 to 6,240,257 conidia/mL diet (Figure 2C). That is, *M. anisopliae* was 11.2 times more pathogenic for the viruliferous whitefly than it was for the non-viruliferous whitefly.

### 3.2. Discrimination between Bacterial Composition in Non-Viruliferous and Viruliferous Whitefly

#### 3.2.1. Overview of Microbiotas in Whitefly Fed on *M. anisopliae*

In order to assess if *M. anisopliae* can affect whitefly microbiota composition, we supplemented the diet with *M. anisopliae* various LC_20_, LC_30_, and LC_50_ of whiteflies separately. Microbial communities in whitefly were determined through 16S rRNA sequencing. Totally, 2,205,378 sequences and 389 OTUs (Phylum: 26, Class: 50, Order: 108, Family: 173, Genus: 256, Species: 339) were identified from whitefly samples with an average length of 415 bp. Good’s coverage estimates indicated that over 99% of the species diversity was captured in the 48 samples (Appendix A), signifying that the sampling depth was adequate. Rarefaction curves, nearing a saturation plateau, exhibited substantial variability in OTU numbers across samples, with higher OTU density observed in the upper layer compared to lower strata. Furthermore, these curves suggested an impact of *M. anisopliae* on the richness of the whitefly community (Appendix A).

#### 3.2.2. Diversity Comparison of Bacteria in Non-Viruliferous and Viruliferous Whitefly

The bacterial communities between non-viruliferous and viruliferous whiteflies were compared, considering the effects of *M. anisopliae* exposure. Analysis of alpha diversity metrics showed that when exposed to the fungus, bacterial richness (ACE and Obs indices) was significantly higher in viruliferous versus non-viruliferous whiteflies, while diversity (Shannon and Simpson indexes) did not differ significantly between groups (Figure 3, Appendix A). In control samples, viruliferous whiteflies exhibited significantly higher ACE richness but only slightly (non-significantly) lower Shannon diversity compared to non-viruliferous whiteflies (Figure 3A,C). After fungus exposure at LC_20_, non-viruliferous whiteflies displayed small, non-significant decreases in richness and increases in diversity (Figure 3B,D). By contrast, viruliferous whiteflies showed modest increases in both richness and diversity, albeit statistically non-significant. Furthermore, CCYV presence itself (in control groups) appeared to reduce microbial richness and diversity in whiteflies (Figure 3A,C). Together, these results indicate that both *M. anisopliae* exposure and CCYV infection can influence metrics of alpha diversity associated with the whitefly microbiota. While some metrics differed significantly between viruliferous and non-viruliferous groups, changes within groups following fungal exposure were relatively modest and often not statistically significant. Further investigation into the biological relevance of these small shifts is merited in future studies.

Dendrograms (tree diagrams) were generated to illustrate the relatedness and composition of bacterial communities from different groups of whiteflies in the study. The cluster analysis dendrogram showed three main branches (Figure 4A). The first branch consisted mostly of viruliferous (CCYV-carrying) whiteflies, primarily from the untreated control group. The second branch contained viruliferous whiteflies that had been exposed to the fungus *M. anisopliae* at a certain concentration (LC_50_). The third branch split into two smaller sub-branches, one comprised of viruliferous whiteflies and the other of non-viruliferous whiteflies. It was noted that whiteflies treated with the same concentration of *M. anisopliae* exhibited a tendency to cluster more closely together in the diagram. In general, the distances of samples in the same branch were within 0.3 of each other (a value of 0 means they have the same composition, and a value of 1 means they do not share any species). This finding suggests that both *M. anisopliae* exposure and CCYV presence affect the similarity of bacterial communities residing within whiteflies.

The PCoA of Bray-Curtis dissimilarity indexes further showed a statistically significant distinction in the microbial composition of viruliferous and non-viruliferous whitefly fed with *M. anisopliae*-infected leaf (*R*^2^ = 0.3028, *p* = 0.001, by ADONIS) with PCoA1 (56.25%) and PCoA2 (29.26%) explaining 85.51% of the variation (Figure 4B). Substantial differences in the bacterial profiles of the whitefly were substantiated through nonmetric multidimensional scaling (NMDS) analyses, as illustrated in Figure 3C. These analyses demonstrated that *M. anisopliae* and CCYV induced distinct alterations in the microbial structure, evident from the clustering of samples in the plots (Figure 4).

#### 3.2.3. Composition of Whitefly Bacterial Community

Taxonomic analysis across all samples revealed that Proteobacteria constituted the dominant phylum. The three most abundant phyla identified were Proteobacteria, Bacteroidota, and Actinobacteria, as depicted in Figure 5A. Notable differences in relative abundance were observed between non-viruliferous and viruliferous whiteflies, with significant variations in bacterial composition among the groups. Specifically, the Proteobacteria population was markedly more abundant in the control group of viruliferous whiteflies (99.17%)*,* and the *M. anisopliae* exposed group in viruliferous whiteflies (>96.61%). At the same time, it decreased sharply in the non-viruliferous whitefly (97.28%), especially when it was exposed to *M. anisopliae* (<97.28%). The prevalence of Bacteroidota in the control group of viruliferous whitefly was 0.33%. Conversely, a significant increase was observed in those exposed to *M. anisopliae*, with rates of 1.05%, 0.89%, and 1.13%, as viruliferous whitefly showed the same trend in both concentrations (1.14%, 1.82%, 1.00%, 1.22%). The Actinobacteriota reached a minimum abundance in the control group viruliferous whitefly (0.11%), which increased when exposed to *M. anisopliae* (0.19%, 0.14%, 0.43%). Meanwhile, *M. anisopliae* showed no significant effect on non-viruliferous whitefly (around 1.51%).

A heatmap was produced to elucidate the spatial distribution of various OTUs across distinct groups (Figure 5C). This heatmap, delineating the 50 most abundant genera, provided an exhaustive depiction of the bacterial community structure. Certain bacterial taxa demonstrated a persistent colonization pattern in whiteflies, including *Rickettsia*, *Portiera*, *Hamiltonella*, and *Curtobacterium* (Figure 5B,C). Viruliferous whitefly exposed to *M. anisopliae* at LC_50_ exhibited a composition of bacterial taxa that was analogous to non-viruliferous whitefly, whereas the abundance in non-viruliferous whitefly exposed to *M. anisopliae* at LC_20_ appeared to be similar to LC_30_. *Rickettsia* emerged as the predominant bacterial genus within the whitefly population (>51.78%); nonetheless, it significantly decreased in the control group of viruliferous whitefly (42.92%). In contrast, *Portiera* exhibited the greatest species richness in the control group of viruliferous whiteflies (47.53%). *Hamiltonella*, *Cardinium*, and *Curtobacterium* also constituted a principal microbial component of non-viruliferous whiteflies (>9.51%, >0.9, >1.27%) but comprised a lesser portion of viruliferous whiteflies (<9.8%, <1.03%, <1.21%).

In addition, the linear discriminant analysis (LDA) effect size (LEfSe) algorithm was used to identify significantly enriched taxa within the non-viruliferous and viruliferous whitefly population. This analysis revealed a divergence in 613 taxa, distinguished by marked variations in their relative abundances. A comprehensive evaluation indicated that *Hamiltonella* demonstrated a pronounced enrichment in the non-viruliferous whitefly exposed to *M. anisopliae* at LC_20_ concentrations. Conversely, *Rickettsia* had the highest relative abundance in that of LC_30_ and LC_50_ concentrations. *Portiera* had the highest relative abundance in the control group of viruliferous whiteflies (Figure 5D,E and Appendix A). These shifts in microbial community composition appeared to be intricately linked with variations in *M. anisopliae* exposure and CCYV presence, suggesting that *M. anisopliae* is instrumental in delineating the microbial community architecture among these whitefly populations.

## 4. Discussion

The whitefly, often termed a “super pest”, has become a significant subject of research due to concerns surrounding its resistance mechanisms [27,28]. This insect’s rapid adaptability and resistance to predominant insecticide classifications pose a formidable threat to global agricultural stability [29]. Although insecticide resistance is a ubiquitous issue, undermining the efficacy of pest management worldwide, limited scholarly attention has been allocated to deciphering the dynamics between insect vector-mediated transmission of plant viruses and the resistance to insecticides [12,30].

The fungus *M. anisopliae* is an important biological control agent for insects. It also helps scientists understand insect-pathogen interactions [18]. Some plant viruses exclusively transmitted by whiteflies facilitate major outbreaks [31]. These viruses reprogram plant defenses to benefit whiteflies but harm non-vectors [5]. For instance, the CCYV makes whiteflies more tolerant to the insecticide flupyradifurone. Notably, viruliferous whiteflies exhibit a markedly elevated LC_50_ relative to their non-infected counterparts [32]. Considering the semi-persistent transmission of CCYV, it can be hypothesized that resistant whiteflies have the potential to disseminate the virus to alternate host plants at a swifter rate compared to their sensitive counterparts. Recent theoretical advancements suggest that CCYV bolsters the capacity of imidacloprid-sensitive whiteflies to endure external adversities, specifically insecticides [12].

In the present investigation, we elucidated several key findings: (1) *M. anisopliae* exhibits pronounced larvicidal activity, with viruliferous whiteflies manifesting a significantly heightened sensitivity compared to their non-viruliferous counterparts; (2) the larvicidal potency of *M. anisopliae* is contingent upon both concentration and the viral infection status of the whitefly, and (3) *M. anisopliae* exerts a detrimental impact on the microbial community structure within the whitefly.

Interestingly, our research highlighted the increased sensitivity of whiteflies to *M. anisopliae* when influenced by viral triggers. It is significant to note that the CCYV infection augments the fitness of the whitefly when not exposed to *M. anisopliae*, as evidenced by an elevated survival rate and a diminished LT_50_. However, when exposed to *M. anisopliae*, the fitness of the whitefly decreases. This discovery stands as a potentially pioneering observation concerning the increased susceptibility of a viruliferous vector to an insecticidal agent. Furthermore, there is a clear positive correlation between the mortality rate and the concentration of *M. anisopliae*.

Insecticides can exert an indirect influence over insect-vectored plant diseases by modulating vector densities. It is well-documented that plant viruses can markedly alter whitefly host preferences and feeding behaviors, either directly or indirectly, through plant-mediated mechanisms [33]. Conventionally, non-viruliferous whiteflies exhibit a proclivity for virus-infested plants, whereas their viruliferous counterparts are predisposed to select uninfected plants [34,35]. The foundational strategy for the mitigation of plant viruses pivots on effective vector management [36]. The CCYV infection potentiates the susceptibility of whiteflies to *M. anisopliae*, suggesting that leveraging this biological mechanism could be instrumental in curtailing or obliterating viral transmission.

The findings from our investigation into the effects of *M. anisopliae* on whiteflies reveal a particularly significant aspect of its potential for pest control: viruliferous whiteflies exhibit markedly increased susceptibility to this fungal pathogen compared to non-viruliferous whiteflies. This differential susceptibility underscores the potential of *M. anisopliae* to serve as a critical tool in managing populations of whiteflies that are vectors for plant viruses, thereby providing a strategic advantage in controlling the spread of these pathogens in agricultural settings. Incorporating *M. anisopliae* into integrated pest management programs offers a promising avenue for enhancing pest control strategies while mitigating the adverse impacts associated with chemical pesticides. Given the heightened sensitivity of viruliferous whiteflies to *M. anisopliae*, targeted applications could be particularly effective during outbreaks of plant viral diseases. This strategy not only helps in directly reducing the vector population but also minimizes the transmission of viruses, potentially leading to healthier crops and reduced economic losses [15]. Effective integration of *M. anisopliae* into IPM should be adaptive and responsive to real-time monitoring data of pest and viral prevalence, ensuring that the applications are both timely and context-specific.

There is growing evidence that symbiotic bacteria can manipulate insect vectors [37]. These manipulations can directly or indirectly impact vector behavior and the complex interactions between insect hosts and the plants they feed on [38,39]. The successful establishment of an infection in an insect vector relies on intricate virus-bacteria interactions [40,41]. Moreover, there have been instances wherein symbiotic entities mediated alterations in insecticide susceptibility [42]. Interestingly, some symbionts can hijack vector behavior to enhance plant virus transmission. Regarding whiteflies, their facultative symbionts appear to serve diverse functions, notably conferring insecticide resistance [43].

A previous study demonstrated the collaborative role of horizontally acquired bacterial genes in tandem with *Portiera* in the production of pantothenate. This cooperation is instrumental in synchronizing the fitness trajectories of whiteflies and their symbionts [44]. Our findings seem to be consistent with these insights: as the relative abundance of *Portiera* increases, the LT_50_ is elongated (Figure 4 and Appendix A). *Portiera* provides whiteflies with adaptive responses to environmental fluctuations [45]. In contrast, infections with *Rickettsia* appear to amplify whiteflies’ adaptive responses to new environments, thereby magnifying their ecological impacts [46]. Intriguingly, increased susceptibility to insecticides among whiteflies has been linked to an increased relative density of *Rickettsia* [47]. Nonetheless, Pan et al. observed that thiamethoxam-resistant whitefly populations were characterized by a greater abundance of *Rickettsia* but reduced presence of *Portiera* and *Hamiltonella* symbionts [48]. Moreover, *Rickettsia* has been observed to confer a suppression effect on the quantity, proliferation, and sporulation of pathogen infections, potentially functioning as a significant barrier against the further transmission of the pathogen [37]. Our results revealed that *M. anisopliae*-resistant whitefly populations hosted a more prolific presence of *Rickettsia* and *Hamiltonella* but were deficient in *Portiera*. Moreover, *Hamiltonella* not only supports the fitness of *Sitobion miscanthi* but also directs its host toward increased insecticide resistance upon exposure [49]. Given its capacity to synthesize a suite of five B vitamins, it is conceivable that entities such as biotin may aid *Hamiltonella* in inhibiting autophagy, thereby ensuring its persistence within bacteriocytes [50]. It is worth noting that viruliferous whiteflies exhibited a heightened vulnerability to *M. anisopliae*, possibly attributable to their reduced abundance of *Hamiltonella* compared to their non-viruliferous counterparts. Such varied symbiont behaviors could be a culmination of factors ranging from the specific symbiont species or strains, the genetic constitution of the host, or the interplay with co-inhabiting symbionts [51].

One of the pivotal determinants governing the evolutionary trajectories and extended phenotypes of insects is their intricate symbiotic associations with various microbes [52]. Disrupting these relationships can substantially impact mutualistic insect-plant interactions. Prior studies reveal a nuanced “trade-off” between virus transmission ability and insecticide resistance in insects [12].

Our study sheds light on how *M. anisopliae* impacts the microbial communities within whiteflies. Such microbial shifts subsequently modulate the vector’s proficiency in virus transmission and its sensitivity thresholds to insecticidal agents. Understanding these microbial interactions provides a deeper insight into the mechanism through which *M. anisopliae* exerts its effects, suggesting that changes in the microbiota could be leveraged to enhance biocontrol efficacy. Future research will focus on dissecting these interactions to better predict and enhance the impact of microbial-based control strategies on pest populations.

## 5. Conclusions

In this study, we investigated the effects of the entomopathogenic fungus *M. anisopliae* on both viruliferous and non-viruliferous whiteflies, uncovering significant findings that enhance our understanding of biological control mechanisms and their potential applications in IPM. Our research not only demonstrated the increased susceptibility of viruliferous whiteflies to *M. anisopliae* but also revealed the substantial impact of fungal exposure on the microbial communities within whiteflies.

The alteration in the microbiota of whiteflies upon exposure to *M. anisopliae* points to potential mechanisms through which this fungus exerts its effects. These alterations may weaken the whiteflies by disrupting their nutritional and physiological homeostasis, further enhancing the biocontrol efficacy of *M. anisopliae*. Understanding these microbial interactions opens new avenues for biopesticide development, where combinations of microbial agents could be used synergistically.

Further investigation into the long-term effects of *M. anisopliae* on whitefly populations and their associated plant viruses will yield significant insights. Additionally, research into the genetic basis of the interaction between whiteflies, *M. anisopliae,* and their microbiota could provide deeper insights into the molecular mechanisms that underpin these interactions. This knowledge could lead to the development of genetically enhanced strains of *M. anisopliae* with improved biocontrol capabilities.

## Figures and Tables

**Figure 1 microorganisms-12-00887-f001:**
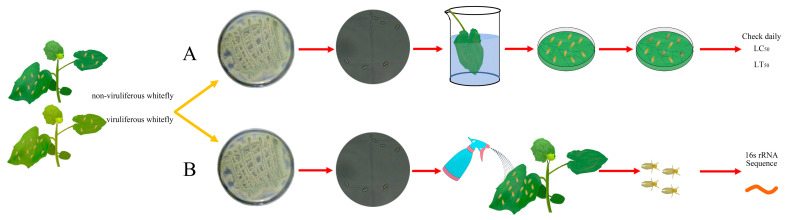
Flow Diagram of Experiment. (**A**) *M. anisopliae* bioassay of non-viruliferous and viruliferous whiteflies, (**B**) Effect of *M. anisopliae* on the Symbiotic bacteria to non-viruliferous and viruliferous whiteflies.

**Figure 2 microorganisms-12-00887-f002:**
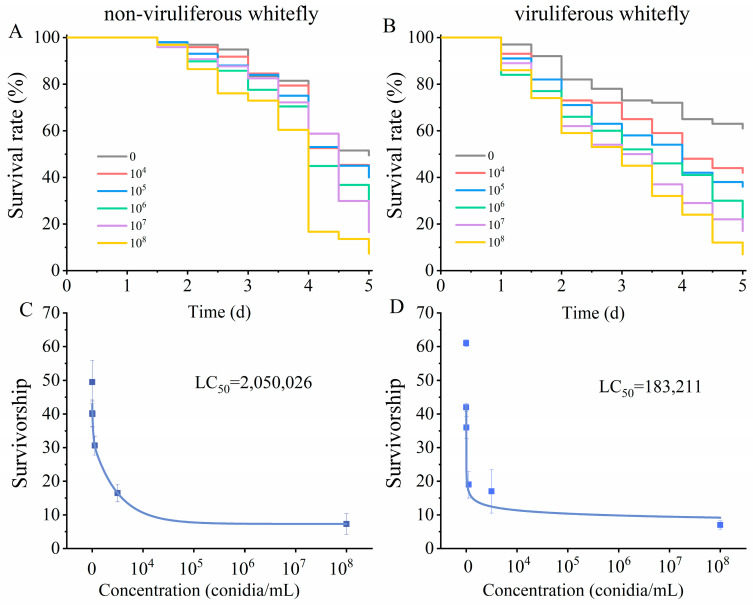
Direct toxicity and LC_50_ of *M. anisopliae* at concentrations on the survival of whitefly feeding bioassays. The survival rates of non-viruliferous (**A**) and viruliferous (**B**) whitefly exposed to *M. anisopliae*. LC_50_ of *M. anisopliae* in non-viruliferous (**C**) and viruliferous (**D**). Values at each concentration represent the mean of 5 biological replicates ± SEM. Graphs for *M. anisopliae* refer to fitted values based on quadratic logistic regression. LC_50_ represents lethal concentration, causing 50% mortality after 5 days.

**Figure 3 microorganisms-12-00887-f003:**
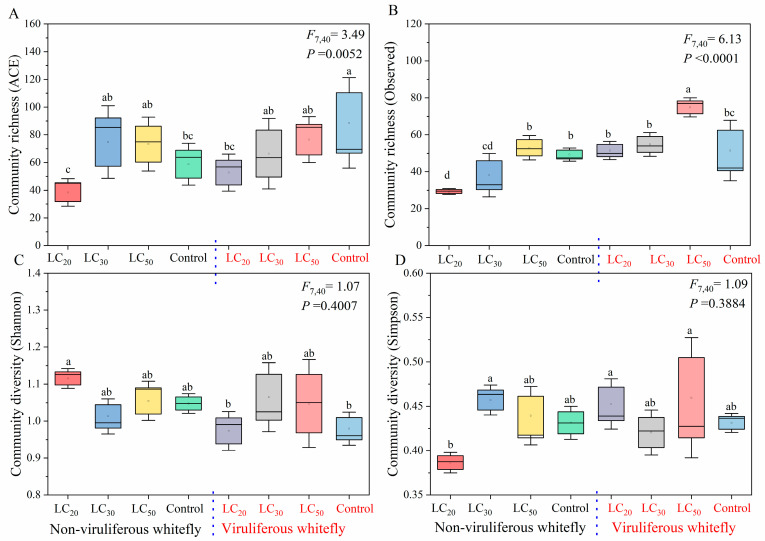
Boxplot of species richness and community diversity of the bacterial communities in whitefly. Alpha diversity plots were measured with ACE (**A**), Obs (**B**), Shannon (**C**), and Simpson (**D**) indexes of samples. Different lowercase labels above each group indicate significant differences (one-way ANOVA, LSD post hoc test, *p* < 0.05) in group mean value. N means non-viruliferous whitefly, V means viruliferous whitefly, LC20, LC30, and LC50 refer to the corresponding Lethal Concentration 20/30/50 concentration of *M. anisopliae*, respectively, and C means control group.

**Figure 4 microorganisms-12-00887-f004:**
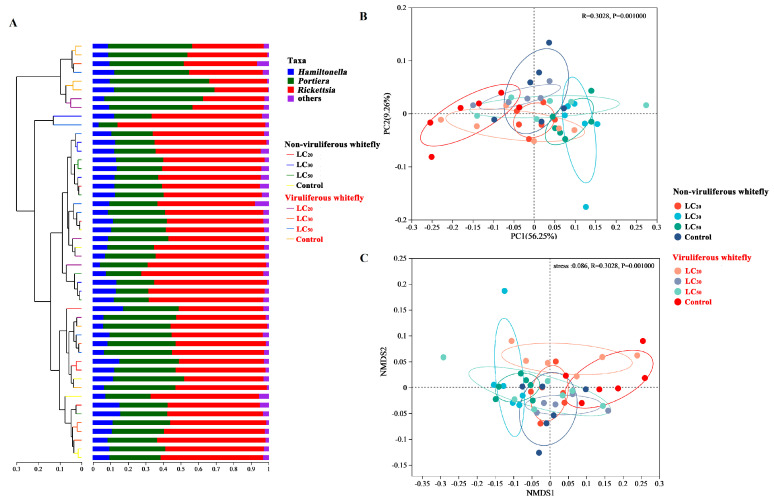
Bacterial communities in the non-viruliferous and viruliferous whitefly after exposure to *M. anisopliae*. (**A**) Unweighted pair-group method with arithmetic means (UPGMA) analysis of microbial community structure based on 16S rRNA gene amplicon sequencing data. (**B**,**C**) Principal-coordinate analysis (PCoA) and nonmetric multidimensional scaling (NMDS) of bacterial communities based on Bray-Curtis dissimilarities. The proportion of variance elucidated by the PCoA and NMDS axes is denoted in parentheses.

**Figure 5 microorganisms-12-00887-f005:**
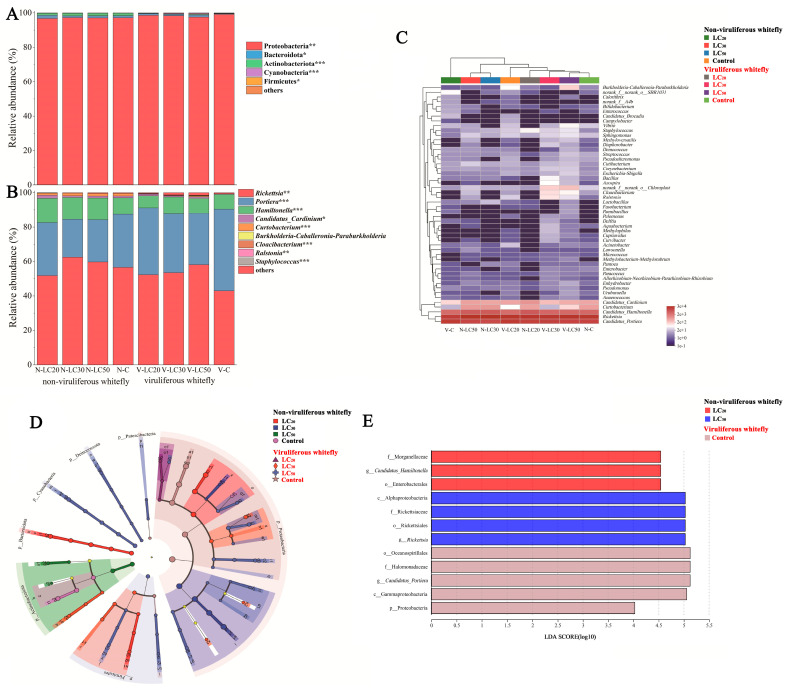
Effects of *M. anisopliae* on the bacterial community compositions of the non-viruliferous and viruliferous whitefly. Bacterial composition of the different communities at the phylum level (**A**) and genus level (**B**). (Non-parametric Kruskal–Wallis test * 0.01 < *p* ≤ 0.05, ** 0.001 < *p* ≤ 0.01, and *** *p* ≤ 0.001). (**C**) Heat map of major taxa over the whitefly at the genus level. Cluster analysis using the Bray–Curtis distance and the complete-linkage method. Each column represents a single replicate of each of the seven treatments. Columns were clustered according to the similarity of bacterial abundance profiles. Each row represents an OTU assigned to the genus level. Color gradient represents the proportion of species. The plotting scale, from red to blue, indicates the decrease in the richness of bacterial communities. A cladogram generated by LEfSe analysis shows enriched taxa in whiteflies from non-viruliferous and viruliferous whiteflies with LDA scores of >4 (**D**,**E**).

## Data Availability

The raw reads were submitted to the NCBI Sequence Read Archive (SRA) database with an accession number PRJNA1019088.

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
