# Peer review of "Managing Super Pests: Interplay between Pathogens and Symbionts Informs Biocontrol of Whiteflies"

_microorganisms, 2024, doi:10.3390/microorganisms12050887_

Round 1
Reviewer 1 Report
Comments and Suggestions for Authors
The manuscript addresses an important topic within the context of whitefly management, as a biological control tool. I suggest that others rewrite the methodology in order to schematize how the experiments were carried out. Improve the resolution of the images in figure 4. The discussion can include points about the applicability of the results obtained in the work. for example, how can these findings be applied to the management of insects associated with the use of Metharhizum?
Author Response
Dear Reviewer,
Thank you for your valuable feedback and suggestions concerning our manuscript. We appreciate your recognition of the importance of our topic within the context of whitefly management using biological control tools. We have carefully considered each of your comments and have made the following revisions to our manuscript:
(1) Methodology Revision: As suggested, we have rewritten the methodology section to provide a clearer schematic representation of the experimental procedures. We have included flowcharts and step-by-step descriptions to enhance clarity and ensure that the experiments can be easily replicated by others in the field.
(2) Improvement of Image Resolution in Figure 4: We have replaced the previous images in Figure 4 with higher-resolution ones to ensure that the details are clear and effectively communicated. This should facilitate better understanding and analysis of the data presented.
(3) Discussion on Applicability of Results: We have expanded the discussion section to include detailed points on how the findings from our study can be applied in the practical management of insects, specifically with the use of Metharhizium. We now explore potential implications for both small-scale and large-scale agricultural settings, providing a framework for how our results can inform future biological control strategies.
We believe these changes address your concerns and strengthen the manuscript significantly. We hope that our revisions meet your expectations and look forward to your further suggestions.
Reviewer 2 Report
Comments and Suggestions for Authors
The paper "Managing Super Pests: Interplay Between Pathogens and Symbionts Informs Biocontrol of Whiteflies" contains interesting data which look properly analyzed. Some notes are reported on the attached document

Minor editing of English language required
Author Response
Dear reviewer,
Thank you for your insightful feedback on our manuscript. We appreciate your positive remarks on the data presented and the analysis conducted. We have carefully reviewed the notes provided in the attached document and have addressed each comment.

Reviewer 3 Report
Comments and Suggestions for Authors
Dear authors,
The use of entomopathogenic fungi (EF), such as Metarhizium anisopliae, is a practice that has been widely used in agriculture worldwide. EF are products with a broad spectrum of action and are used to control Bemicia tabaci. For this reason, I disagree that the impacts of M. anisopliae on B. tabaci are little known, with there being hundreds of articles that deal with this subject. However, something that aroused my curiosity is linked to symbiotic microorganisms of B. tabaci. In Diaphorina citri (Hemiptera: Liviidae), these symbionts can help the insect in stressful situations. Furthermore, D. citri infected with the bacterium Candidatus liberbacter may have a greater tolerance to synthetic insecticides. In this context, I would like the authors to reflect on de-emphasizing M. anisopliae and valuing more information about symbiotic microorganisms. Therefore, seeking to contribute to the text, here are some observations:
The introduction must be revised, as there is little support that the authors give to their readers to support the hypothesis that M. anisopliae negatively affects the fitness of whiteflies. For example, the authors spend an entire paragraph (line 56-67) on the effects of the abusive use of synthetic insecticides and the benefits of bioinsecticides formulated from EF. As the authors work with sublethal concentrations (LC20, LC30 and LC50), they could introduce their readers to sublethal effects. This could perhaps justify the hypothesis that M. anisopliae negatively affects whitefly fitness. I think it is important to focus on the insect's microbiota, as we have methodology, results and discussion about this, but the introduction is deficient. Therefore, I recommend restructuring the introduction.
The topic material and methods are robust and the methodologies widely used in the literature. However, the authors can be more detailed in topic 2.1. I recommend explaining in more detail how the plants, insects, viruses and fungi were maintained and obtained. I would like to invite authors to reflect on the bioassays described in topic 2.2. and 2.3. In addition to questioning whether the insects used in the bioassays were the same age, does the use of immersion (2.2.) and spraying (2.3.) influence the results?
The results topic is appropriate to the methodology used. However, I recommend that authors provide more information. For example, a lethal time of 3.4 days for non-viruliferous insects and 2.7 days for viruliferous insects (L 163), inform statistical parameters such as chi-square value, standard deviation and confidence interval. This must also be done for lethal concentration. In Figures 1C and 1D, what are the values of LC50 = 2050.026? Conidia per mL? Furthermore, the manuscript must be revised, as there are words written inappropriately (e.g. M. anisopliae l 184).
The discussion topic, as well as the introduction, should be reviewed in order to respond to the results found by the authors.
Author Response
Dear reviewer,
Thank you for your insightful comments and suggestions regarding our manuscript. We appreciate your points on the need to enhance the introduction and provide more detail in our methods and results sections, as well as the emphasis on the role of symbiotic microorganisms in Bemisia tabaci. Here is our response to the issues you raised:
- Introduction Revision: We acknowledge the need to better substantiate our hypothesis regarding the negative impacts of Metarhizium anisopliae on whitefly fitness. We will revise the introduction to include a more thorough review of literature on the sublethal effects of entomopathogenic fungi (EF), specifically how these effects could potentially impair the physiological and reproductive capabilities of tabaci. This revision will also align the narrative with the broad spectrum of action of EF and its implications on whitefly populations.
- Materials and Methods Details: Your suggestion to elaborate on the methods used for maintaining and obtaining plants, insects, viruses, and fungi is well taken. We will expand section 2.1 to include specific details about the sources and maintenance conditions of our experimental organisms. Additionally, we will consider your valuable point regarding the age of insects used in bioassays and the potential influence of different application methods (immersion vs. spraying) on the outcomes.
Uniform Age of Insects: In our experiments, we used whiteflies of the same age to ensure uniformity in susceptibility to Metarhizium anisopliae in both Immersion bioassay and Spraying bioassay. This standardization is crucial for minimizing variability due to age-related differences in immune response or physical condition, thereby providing reliable and replicable results.
Purpose of the Bioassays: The bioassays, as outlined in topics 2.2 and 2.3, were specifically designed to test the effectiveness of the Metarhizium anisopliae against both virally infected and healthy whiteflies. By assessing the lethal median time (LT50), we aimed to quantitatively determine how quickly the fungus can control whitefly populations under different conditions. This measure provides a clear indicator of the fungus's potential as a biocontrol agent.
Immersion bioassay: This method involved directly immersing the whiteflies in a fungal suspension. This approach ensures that each insect is exposed to a controlled, quantifiable amount of the fungal agent, allowing for precise measurement of fungal efficacy. This technique is particularly useful for evaluating direct fungal infection and penetration efficiency under laboratory conditions.
Spraying bioassay: Spraying was used to mimic a more natural application method, as it reflects how fungal biocontrol agents are typically applied in agricultural settings. This method tests the fungus’s ability to infect whiteflies in conditions that are more representative of those in the field, including variables such as spray coverage and environmental effects on fungal viability.
Both methods are crucial for thoroughly evaluating the potential of the green muscardine fungus as a biocontrol agent in diverse agricultural environments. By employing both immersion and spraying techniques, the research can provide insights into the performance of the fungus under controlled conditions as well as its practical effectiveness and feasibility in field applications.
- Enhanced Reporting in Results: We have provided additional statistical details as suggested, including chi-square values, standard deviations, and confidence intervals to support our findings more robustly. We realize the necessity of being more explicit about our lethal concentration data, including precise units of measurement (e.g., conidia per mL for LC50 values), to enhance the clarity and reliability of our data presentation.
- Discussion and Manuscript Revision: We agree that the discussion should more effectively tie back to the results presented. We have revised this section to better reflect the findings and integrate discussions on how M. anisopliae's impact on the whitefly microbiota could influence overall pest management strategies. We have corrected inappropriate terminologies to ensure that the manuscript meets the high standards of scientific communication.
Focus on Symbiotic Microorganisms: We agree on your suggestion to place more emphasis on the role of symbiotic microorganisms in the ecological fitness and stress response of B. tabaci compelling. We have improve our discussion on how these symbionts could modify host responses to both biotic and abiotic stresses, potentially influencing the effectiveness of biocontrol agents like M. anisopliae.
Your feedback is instrumental in helping us improve our manuscript. We are committed to addressing these points thoroughly in our revisions to ensure our research contributes valuable insights into the sustainable management of whitefly using biocontrol strategies.
Thank you for all above comments and suggestions for helping us refine our work.
Round 2
Reviewer 3 Report
Comments and Suggestions for Authors
Dear authors,
In my opinion, the manuscript has undergone substantial improvement. I just ask for one last review, as there are scientific names written incorrectly. Congratulations on the work.
Sincerely
Author Response
Dear Reviewer,
Thank you for your feedback. I appreciate the recognition of the improvements made to the manuscript. We had conducted a thorough review to correct the inaccuracies in the scientific names as suggested. Thank you for your valuable input.